# Stakeholder-Informed Hydroclimate Scenario Modeling in the Lower Santa Cruz River Basin for Water Resource Management

Neha Gupta [1,*], Lindsay Bearup [2], Katharine Jacobs [3], Eve Halper [4], Chris Castro [5], Hsin-I Chang [5] and Julia Fonseca [6]

1  Arizona Institute for Resilience, University of Arizona, Tucson, AZ 85719, USA
2  Water, Environmental and Ecosystems Division, Bureau of Reclamation, Denver, CO 80215, USA; lbearup@usbr.gov
3  Center for Climate Adaptation Science and Solutions, University of Arizona, Tucson, AZ 85719, USA; jacobsk@arizona.edu
4  Phoenix Area Office, Bureau of Reclamation, Glendale, AZ 85306, USA; ehalper@usbr.gov
5  Department of Hydrology and Atmospheric Sciences, University of Arizona, Tucson, AZ 85719, USA; clcastro@arizona.edu (C.C.); hchang05@arizona.edu (H.-I.C.)
6  Madrean Resources LLC, Tucson, AZ 85716, USA
*  Correspondence: nehagupta@arizona.edu

**Abstract:** The Lower Santa Cruz River Basin Study (LSCRB Study) is a collaborative effort of regional and statewide water management stakeholders working with the US Bureau of Reclamation under the auspices of the 2009 SECURE Water Act. The impacts of climate change, land use, and population growth on projected water supply in the LSCRB were evaluated to (1) identify projected water supply and demand imbalances and (2) develop adaptation strategies to proactively respond over the next 40 years. A multi-step hydroclimate modeling and risk assessment process was conducted to assess a range of futures in terms of temperature, precipitation, runoff, soil moisture, and evapotranspiration, with a particular focus on implications for ecosystem health. Key hydroclimate modeling process decisions were informed by ongoing multi-stakeholder engagement. To incorporate the region's highly variable precipitation pattern, the study used a numerical "weather generator" to develop ensembles of precipitation and temperature time series for input to surface hydrology modeling efforts. Hydroclimate modeling outcomes consistently included increasing temperatures, and generated information related to precipitation responses (season length and timing, precipitation amount) considered useful for evaluating potential ecosystem impacts. A range of risks was identified using the hydroclimate modeling outputs that allowed for development of potential adaptation strategies.

**Keywords:** hydroclimate modeling; climate change; water resource management; riparian health; stakeholder-informed modeling; weather generator; Santa Cruz River; Tucson

## 1. Introduction

Shifting climate conditions in combination with land use changes due to population growth create significant challenges for water and environmental resource efforts [1,2]. Resource managers in semi-arid regions are currently faced with many competing demands for a limited water supply and with limited information. Often, the information that is available to support ecosystem management is experience-based and therefore it is useful to engage practitioners focused on environmental preservation and restoration in strategic planning efforts and in establishing goals of modeling efforts [3,4]. However, understanding how future climate conditions will impact water management is an ongoing challenge. The development of technical information such as hydroclimate projections requires careful attention to the intended use of the information to be developed [4]. There have been increased efforts to link modeling and scenario planning efforts to support on-the-ground practitioners [5–7]; in this paper, we focus in particular on input from

environmental managers and stakeholders who are interested in environmental restoration and preservation. We found that development of place-based hydroclimate scenarios, in partnership with stakeholders, produced useful information to evaluate environmental impacts related to changing water availability under future climate in the Lower Santa Cruz River Basin (LSCRB).

*Co-Production of Hydroclimate Information*

Multiple studies have recognized that successful co-production of knowledge is characterized as involving stakeholders in multiple phases that include problem definition, research question development, conduction of research activities, production of results, and knowledge dissemination [8,9]. The use of regional assessments to produce information that can be used to respond to climate impacts requires applied science that is generated via interaction with stakeholders that can be used to address complex, ongoing problems and stakeholder perceptions of immediate needs for decision-making, resulting in a need to strike a balance between "usable knowledge" and high-quality technical information [8]. In the context of hydroclimate modeling, a string of decisions that range from selection of carbon emission scenarios, global climate models, and downscaling methodologies to development of hydrologic models can highly influence predictions generated by modeling efforts due to model sensitivities [10]. However, the development of hydroclimate scenarios, particularly those that incorporate multiple downscaling methods and the use of a stochastic rainfall generator, such as the "weather generator" detailed in this study, can be complicated and require extensive discussion and potential education of resource managers and stakeholders [11].

There are limited case studies that detail how interaction with stakeholders can transform or influence production of science to affect how scientists formulate research questions and carry out research efforts. The Murray Darling Basin Plan (MDBP) is an effort undertaken in the Murray Darling Basin, located in Australia, that incorporated similar hydroclimate modeling efforts and consideration of stakeholder needs in the context of water management decisions, but did not appear to incorporate stakeholder input in the process of hydroclimate scenario design and considered modeling and stakeholder engagement as two discrete phases [11,12]. A study conducted on the Upper Santa Cruz River (USCR), located upstream of this project's study area (LSCRB), used dynamically downscaled climate projections that were translated into ensembles of rainfall realizations using a stochastic rainfall generator [6]. The USCR study considered riparian impacts via the evaluation of groundwater thresholds that could maintain riparian health for a number of key species in the region in response to stakeholder concerns. The study detailed herein used a similar hydroclimate modeling approach to the USCR study. However, the development and evaluation of adaptation strategies in response to hydroclimate modeling efforts does not appear to have been conducted as part of the USCR study.

The aim of this paper is to describe the series of hydroclimate modeling efforts that were informed by regional stakeholders throughout the process of development and in the production of hydroclimate modeling output as a case study. This study used a regionalized co-production process to integrate local expert knowledge to increase utility of research results and development of adaptation strategies to better anticipate and respond to local climate impacts [9].

This study was conducted in a region in which multiple stakeholder engaged efforts have occurred due to the "frontline" nature of climate impacts that include increasing drought and reduced surface water flows in conjunction with increasing urbanization associated with increasing populations [6,13]. These threats, coupled with the nature of legacy environmental impacts in the region, result in the need to evaluate water resource concerns with consideration to future climate impacts on both human and environmental systems to support regional stakeholder concerns and priorities. The process summarized in this study is part of a larger modeling and engagement effort in which population growth scenarios, hydroclimate scenarios, and groundwater modeling efforts were combined to

evaluate changes in groundwater pumping and to identify of areas of concern via the Lower Santa Cruz River Basin Study (LSCRBS) summarized in Section 2.2. This paper summarizes the portion of the LSCRBS concerned with developing hydroclimate scenarios and associated adaptation strategies that prioritize the addressing of environmental impacts. Technical outcomes of hydroclimate modeling were also incorporated into an online dashboard so stakeholders can continue to use modeling results in adaptation planning efforts in the region, particularly those concerned with evaluating locations for riparian recharge projects.

## 2. Study Methods

### 2.1. Study Area

The study area consists of two deep alluvial basins within the basin and range province of Arizona: the Tucson and Avra/Altar Sub-Basins (Figure 1a). The Tucson Basin is the principal locus of residential and commercial development, with roughly one million residents. The central Avra/Altar Basin is the principal repository for underground municipal storage of imported Colorado River water; the northern portion of the sub-basin includes irrigated agricultural land, but is urbanizing over time.

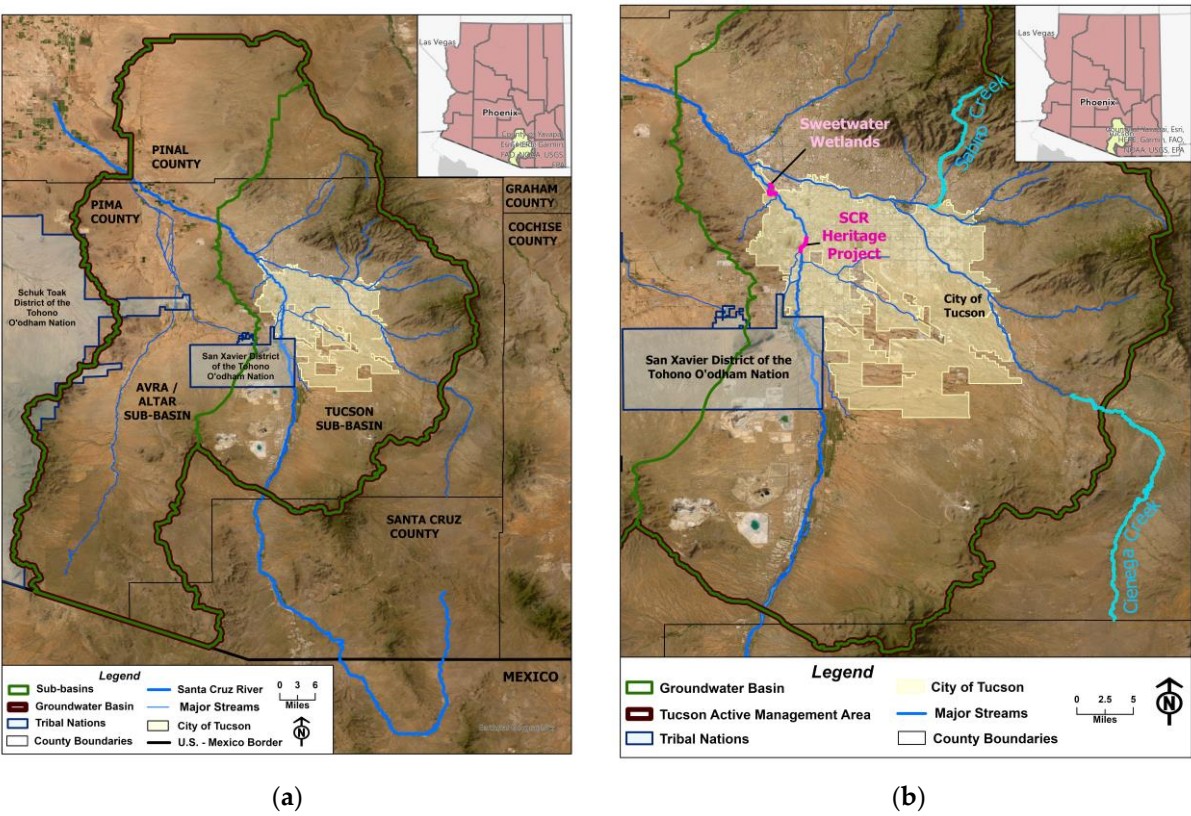

(**a**) (**b**)

**Figure 1.** Map of study area that includes (**a**) key surface water features such as the Santa Cruz River and groundwater basins, and (**b**) closeup of the greater Tucson area with regional waterways and water management projects (reprinted with permission from the US Bureau of Reclamation).

The study area's climate is semi-arid, with both winter and summer rains. The dominant vegetation at lower elevations is desert scrub, with most woodlands restricted to montane areas. The Santa Cruz River originates in Arizona, flowing south into Mexico to the east of the study area, and then turning north. It enters the Tucson Basin in Pima County, and then flows into Avra Valley to the west of Tucson. High mountain ranges to the north and east of the Tucson Basin contribute runoff to the Santa Cruz River via its tributaries and support natural recharge of the aquifer.

The study area boundaries are the same as those of the Tucson Active Management Area, a region managed towards a statutory goal of safe yield by the Arizona Department of Water Resources under the 1980 Groundwater Management Act (Figure 2). The study area includes the service areas of multiple large municipal water providers that were engaged in the study, rangelands and wildlife habitat managed by Pima County, state and federal agencies, and the San Xavier and Schuk Toak Districts of the Tohono O'odham Nation. Large-scale copper mining takes place in the southern portion of the Santa Cruz Sub-Basin; large-scale agriculture occurs in both sub-basins.

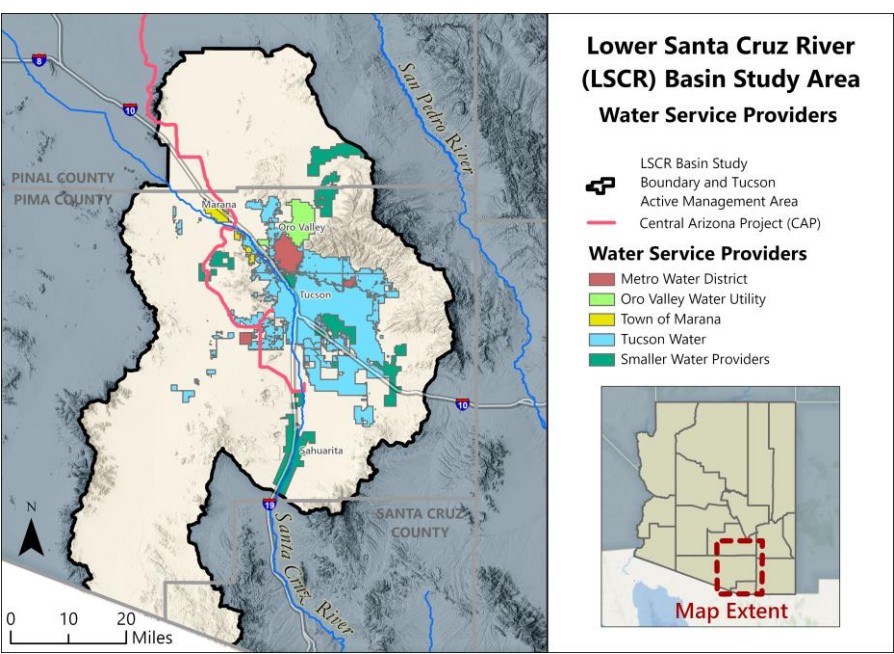

**Figure 2.** Map of boundaries of LSCRBS with water providers included within the study area (reprinted with permission from the US Bureau of Reclamation).

Flowing streams and evapotranspiration from streamside forest and wetlands in central and southern Arizona have been greatly reduced through surface water diversions and groundwater pumping in central and southern Arizona [14–16]. Prior to the Anglo water resource development, portions of the Santa Cruz River flowed year-round, supporting extensive forests of mesquite, cottonwood, and willow, and diverse native aquatic fauna [9]. By the 1940s, perennial surface flows along the Santa Cruz River were eliminated as groundwater pumping lowered the water table, ending a 4000-year history of indigenous agriculture [17] and leading to the elimination of a unique species of pupfish [18]. Further groundwater depletion and land use change reduced riparian vegetation by 75% between 1937 and 2019 along the Santa Cruz River and its tributaries [19] (Figure 1).

Formerly perennial streams in this region now tend to be ephemeral or intermittent and rely on shallow groundwater, stormwater, reclaimed water or other sources to support the riparian ecosystems. It is well-documented that the majority of the biodiversity in Southern Arizona is associated with riparian vegetation [20]. Loss of streamflow has contributed to the listing of certain plants and animals as threatened or endangered under the Endangered Species Act [21] and eventually led to a water rights settlement with the Tohono O'odham Nation that included delivery of imported Colorado River water to the study area. Because of warm spring, summer and fall temperatures and the scarcity of large surface water impoundments, significant recreation occurs in or in proximity to riparian areas (hiking, biking, birdwatching, etc.). Riparian vegetation also contributes to property values in the area [22,23] and supports the productivity of rangelands used for livestock grazing.

Land and water resource managers of the region have growing concerns about the impact of climate change on rangelands, wildlife, and recreation, because a combination of higher temperatures and less reliable precipitation patterns may dramatically impact the viability of existing vegetation and streamflow. Despite the lack of state-level protections, there has been local support for restoring riparian vegetation or streamflow. Local indigenous farming with imported Colorado River water has restored flow to portions of the Santa Cruz River at San Xavier District, south of Tucson. The City of Tucson adopted a "last-on, first off" policy for municipal wells in a portion of the eastern Tucson Basin [24], and the recently completed the Tucson Water Heritage project (releasing water into the Santa Cruz River channel through downtown). The Heritage project and Sweetwater Wetlands are both projects supported by municipal effluent (Figure 1b) [25]. Pima County and its Regional Flood Control District manage over 250,000 acres of land for species covered under a habitat conservation plan, including habitat for the endangered Gila topminnow [21].

### 2.2. Lower Santa Cruz River Basin Study

The Bureau of Reclamation (Reclamation) conducted the LSCRBS in partnership with regional stakeholders across a range of sectors (water providers, industry, agriculture and environment) within the Tucson Active Management Area (https://new.azwater.gov/ama, accessed on 24 March 2023). The 2009 SECURE Water Act authorizes Reclamation to partner with non-federal participants to analyze the impacts of climate change and develop adaptation strategies to meet future water supply and demand imbalances in river basins within the 17 Western United States. The resulting "Basin Studies" promote water supply reliability in the context of climate impacts on both water supply and demand.

Active engagement, partnership, and leadership from a range of actors resulted in numerous innovations in the LSCRBS process to produce relevant, place-based information. This information was used to support a wide range of water resource decisions related to future climate and population growth patterns. The study team also made a dedicated effort to summarize and promote access to technical information in formats that can continue to be useful for a range of future water resource management planning activities, including publicly accessible science summaries and a web-based data retrieval system.

The LSCRBS used a linked series of models to support the development of water demand and supply scenarios and to map geographic areas of concern within the basin. The modeling efforts started with downscaled global climate model (GCM) data for the region, which yielded projected temperature and precipitation information. Subsequent modeling yielded projected changes in streamflow and soil moisture values. These efforts are summarized in Section 2.3.

The LSCRBS is unique among the Reclamation Basin studies to date in that evaluating environmental water supply reliability in the context of climate change was considered a priority. Because of a historic groundwater overdraft and an associated depletion of local streamflows [26,27], and a concern that future changes in temperatures and precipitation patterns could threaten the few remaining riparian ecosystems in the region [28], the study partners agreed on the need to incorporate an examination of water supply impacts for ecosystems into the evaluation. Understanding changes in flow patterns and seasonality are an important focus of land, water resource, and environmental managers, as well as impacts on shallow groundwater levels due to their role in supporting vegetation and influencing water demand across multiple sectors (e.g., outdoor watering demand on municipal water supplies).

### 2.3. Hydroclimate Scenario Development

LSCRBS researchers conducted a series of stakeholder engagement efforts to assess priorities for the study and to develop water supply and demand projections under two different climate scenarios. Use of scenarios allowed for consideration of a range of potential climate and development futures, with strong integration of scientific efforts and participatory approaches to support decision-making under uncertainty [5]. Scenario-

based projections were developed across two time horizons to allow for evaluation of "near future" impacts (2020–2049) and "far future" (2050–2079) impacts.

Hydroclimate modeling efforts are described in detail in US Bureau of Reclamation Technical Memorandum No. ENV-2021-35 [29], and are summarized below where useful to illustrate alternative hydrologic futures, scenario development, and alternative adaptation actions.

### 2.3.1. Climate Scenario Development

Study partners selected climate scenarios to represent a range of future emissions, ultimately narrowing the selection to two "bookend scenarios" labeled "best case" and "worse case" [30,31]. Partners deliberately selected the term "worse case" to emphasize that while this scenario was considered appropriate for high-level planning, it did not encompass all climate-associated risks to the study area; partners were also concerned about the possibility that future emissions could exceed the highest emission scenario available for the study, considered "business as usual." Partners were also aware of 2016 research on the nearby Salt and Verde River basins that demonstrated that dynamically downscaled climate projections produced substantially lower projections of streamflow than the same GCM projections downscaled using the statistical bias-corrected spatially disaggregated method [32]. To avoid an overly optimistic projection of streamflow, the partners requested that a dynamically downscaled climate projection be included in the Study.

Study partners collaborated with local researchers from the University of Arizona Department of Hydrology and Atmospheric Sciences to select from the available dynamically downscaled climate projections. A key criterion in decision-making was the projection's ability to simulate timing of the summer North American Monsoon. After a thorough screening of the available options, Reclamation and the partners agreed to use the Max Planck Institute for Meteorology Earth System Model (MPI-ESM) runs from the Coupled Model Intercomparison Project Phase 5 (CMIP 5) downscaled using the Weather Research and Forecasting Model (WRF). Different generations of GCM climate projections from the Intergovernmental Panel on Climate Change (IPCC) have been reported to have various climate biases over key meteorological variables globally. For the Southwest US, IPCC CMIP5 analyses suggest there will be a delay in North American Monsoon onset in early summer due to increased atmospheric stability but an increase in monsoon precipitation in late summer [33]; CMIP6 projections reasonably capture the precipitation variability but simulate an overly strong North American Monsoon [34]. Even with the latest IPCC GCM projections, the challenge remains to resolve meteorological features over complex, multi-elevation terrain.

To physically simulate weather and climate processes on a regional scale, a regional climate model (RCM) using dynamical downscaling is required. A regional climate model utilizes a global climate model as boundary forcing, then provides enhanced spatial resolution within that boundary that provides improvement in the physical representation of hydroclimate processes. Selection of a GCM for dynamical downscaling requires GCMs with good large-scale forcing mechanisms. The MPI-ESM IPCC projection selected for dynamical downscaling in this project was independently evaluated for performance with respect to the climatological representation of precipitation and temperature and natural climate variability in North America [35]. The dynamically downscaled RCM climate projection used in this manuscript (WRF-MPI) is part of the North American Coordinated Regional Climate Downscaling Experiment (CORDEX-NA). CORDEX-NA is a community effort to dynamically downscale IPCC CMIP5 projections for the North America using regional climate models.

At the time, the only emission scenario, or Representative Concentration Pathway (RCP), available for the dynamically downscaled projections was RCP 8.5, which was selected for use in this study as the "worse case" scenario. To better represent the range of uncertainty related to future emissions, Reclamation technical staff recommended, and partners agreed, to a "best case" that uses the same GCM, run under the RCP 4.5 emissions

scenario and downscaled using the Localized Constructed Analog (LOCA) [36,37] method. LOCA is a statistical downscaling approach, which means it relies on the spatial patterns observed historically to provide more spatial detail to coarse model outputs, in this case outputs from the MPI-ESM. While this statistical downscaling approach assumes that the spatial patterns of the past are relevant in the future, many of the processes controlling these fine-scale patterns, such as topography, are consistent in the future. The LOCA approach also updated the analog-selection method from previous statistical approaches to reduce the effects from the assumption that a historical proxy is available for all future days. This assumption is particularly impactful for extreme events that may not be in the historical record. Subsequently we refer to the WRF-MPI climate projections as the "worse case" and the LOCA-MPI projections as the "best case", based on the emissions scenarios represented by each. Comparing the dynamically downscaled WRF-MPI climate projection against the entire statistically downscaled LOCA dataset, WRF-MPI was found to be well within the range of the projection variation.

### 2.3.2. Seasonality

The evaluation of changes in the seasonality of precipitation was identified as important for management decisions because summer rainfall significantly affects demand for municipal and agricultural water supplies. In addition, timing of rainfall events plays a very significant role in supporting biological productivity and water availability in semi-arid and arid regions, in contrast to wetter regions where long-term average volumes are more likely to be deciding factors in water management decisions [6].

Three seasons were defined for the purpose of this study, the summer "monsoon" season, a "winter wet" season and the "dry foresummer" season (Figure 3). Partners were particularly interested in potential changes to the onset of the monsoon season, which is associated with a drop in municipal water consumption as residents curtail outdoor water use. To determine the beginning of the monsoon, Reclamation staff adopted a dewpoint-based metric formerly used by the National Weather Service that allowed the onset timing to match conditions rather than the static date-based metric currently used. The dewpoint thresholds were selected to match observed historical monsoon-onset timing reported by Ellis et al. (2004) [38] and bias-corrected as needed to ensure appropriate application to the future climate datasets. Monsoon demise (and start of the "winter wet" season) was defined as the day after the last three consecutive days above the same dewpoint threshold.

## Operational Definitions of the Three Seasons

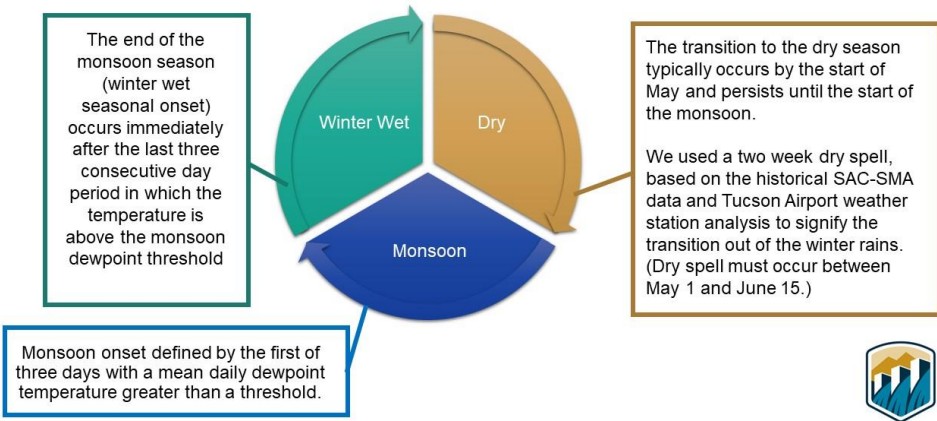

**Figure 3.** Conceptual diagram of precipitation-based seasons with denotation between seasons described in callout boxes. Reprinted with permission from the US Bureau of Reclamation.

The dry season in Tucson is characterized by prolonged periods with low to no rainfall. The transition to this season typically occurs by the start of May and dry conditions persist until the start of the monsoon season. The study used a metric of two weeks without rain to denote the onset of the "dry foresummer", based on historical Sacramento Soil Moisture Accounting (Sac-SMA) model data and Tucson Airport weather station analysis. A daily precipitation threshold of less than 0.01" was used to define rainfall events and maintains consistency with the Southwest Climate Outlook conducted by the Climate Assessment for the Southwest [39–42].

### 2.3.3. Weather Generator

Although the best and worse case climate scenarios bookend a range of future emissions scenarios, they each only provide one possible timeseries of future precipitation and temperature. To capture the uncertainty related to the large variability in local weather from year to year, a weather generator was developed to simulate variability based on historical rainfall and temperature patterns, producing an ensemble of timeseries for temperature and precipitation (Figure 4) [29,43]. The weather generator was used for both climate scenarios ("best" and "worse" cases) across the two planning horizons ("near" and "far" future) of the study.

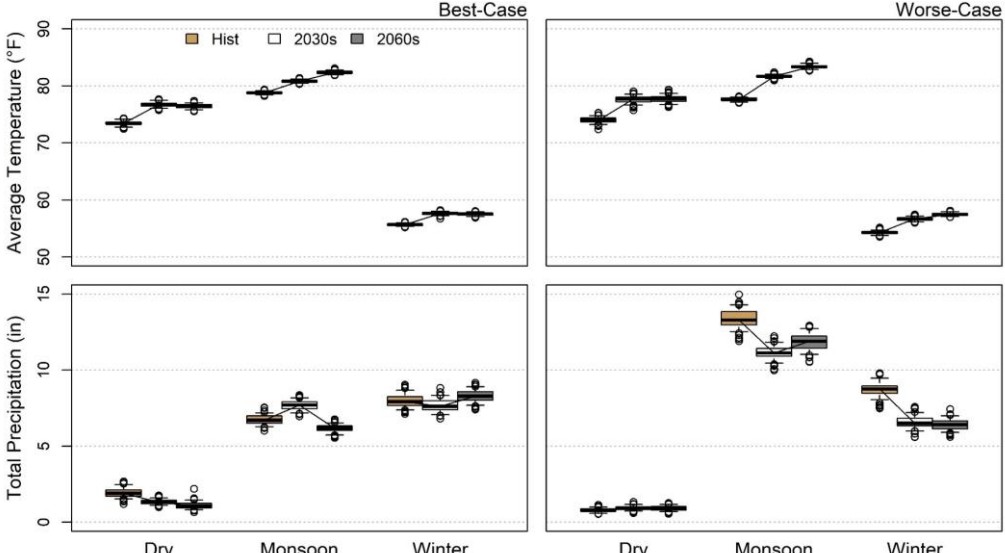

**Figure 4.** Weather generator simulation output presented as 30-year seasonal average across surface water model boundary. Reprinted with permission from the US Bureau of Reclamation.

The weather generator was validated against the historical calibration dataset from the surface water model used in this study and was used to generate 100 daily realizations across sub-basins based on elevation zones for each 30-year period. Daily precipitation occurrence was sampled based on wet/dry day transition probabilities, derived from downscaled climate data, in which "wet" or rainy days were defined as a minimum of 0.01 inches of rainfall. Precipitation was simulated using a nonparametric approach in which previous days' status was used to predict whether daily precipitation occurred. Temperature was simulated using a parametric approach, due to its lower variability relative to precipitation in the region, using an autoregressive linear equation with lag-one persistence and input variables of precipitation occurrence and monthly temperature.

Weather generators can use different statistical approaches to "reshuffle" the deck, and some add additional variability. Here, we chose to rely on the downscaled data to determine the range in magnitude of future storm events, given the studies' efforts to develop future projections. This limits the weather generator to storms available in the input timeseries, which works well for the large range of events in the dynamically-downscaled worse case, but is limited by the lack of large extreme events in the statistically-downscaled best case.

2.3.4. Surface Water Modeling

The Sac-SMA hydrologic model developed by the National Weather Services' Colorado Basin River Forecast Center was used for the surface water modeling portion of the basin study. This model simulates processes related to water movement through the soil column, preserving the water balance, which includes surface runoff, infiltration, interflow, percolation, storage, evapotranspiration, and baseflow. The model is run in a lumped framework, with parameters averaged over elevation zones. Weather generator output, in the form of 100-projection ensembles of projected temperature and precipitation, were input into the Sac-SMA model along with potential evapotranspiration estimates derived using the Hamon method [44], with temperatures predicted from weather generator output that were averaged across the surface water modeling area. The surface water model was calibrated using historical data from 1970–1999.

Surface water modeling results were directly useful for understanding future climate impacts on riparian areas. Projections of streamflow recharge were also used as input into a groundwater model developed by the Arizona Department of Water Resources with updates funded by the Central Arizona Project as part of a separate endeavor. Description of the groundwater modeling efforts is beyond the scope of this paper and is summarized in Bureau of Reclamation Technical Memorandum No. ENV-2021-64 [45].

Sub-basin outputs were presented to stakeholders following development, with attention given to supporting long-term accessibility of hydroclimate modeling data. To promote the use of technical information in a range of future water resources management planning, a web-based GIS tool was developed that presents model outputs by sub-basin in a user-friendly format to support incorporation of information into future prioritization and pilot implementation efforts.

*2.4. Stakeholder-Integrated Water Resources Evaluation*

The primary stakeholders of this effort were the major water users and water interests in the basin, including municipal utilities, and representatives of agricultural, industrial, and environmental sectors. Special effort was made to encourage engagement from mining interests and the two Tribal nations in the study area, the Tohono O'odham and the Pascua Yaqui.

Because the study partners included a large consortium of water interests via the Southern Arizona Water Users Association (SAWUA), the University of Arizona, the Central Arizona Project, Pima Association of Governments, and the Arizona Department of Water Resources, the study had access to a wide array of stakeholder viewpoints and available data compiled and/or developed by regional researchers and practitioners. Project management and engagement efforts were also informed by long-term work on building trusted relationships, designing inclusive engagement strategies, connecting science and decision-making, scenario development, co-production of knowledge, and climate services in water management in the area (e.g., [5,46–48]).

Stakeholder engagement efforts helped inform the development of hydroclimate scenarios as well as the selection of model outputs and summaries (Figure 5). Invited and self-selected stakeholders joined a facilitated process to develop objectives for adaptation, metrics to evaluate adaptation strategies, and requests for technical information to inform adaptation strategy evaluation. The study included an environmental subgroup comprised of representatives from multiple organizations and agencies focused on environmental conservation, water supply, and flood control. The Environmental Subgroup stakeholders participated in multiple engagements (19 meetings over 4 years) that included an internal survey effort that led to the development and refinement of adaptation objectives and information requests from hydroclimate modeling efforts.

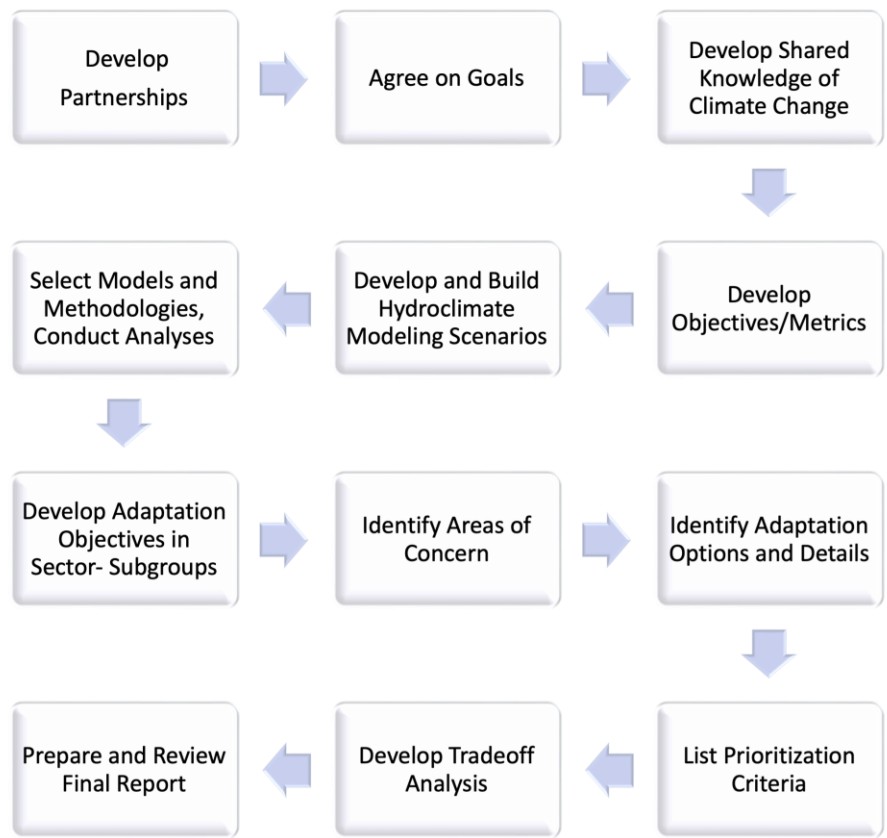

**Figure 5.** Summary of stakeholder engagement process throughout duration of LSCRBS. Stakeholder engagement efforts are summarized in this paper up through the identification of adaptation options and details.

Environmental objectives for adaptation strategies were developed by the environmental subgroup with consideration of multiple stakeholder mandates and concerns including flood control, compliance with existing Arizona Department of Water Resources regulations, and integrated water management efforts. Environmental objectives selected by the subgroup are summarized in Table 1.

**Table 1.** Priority environmental adaptation objectives selected by the environmental subgroup.

| Adaptation Objective | Description |
| --- | --- |
| Enhances or protects high-value habitat | Prioritize consideration of areas that include those with mature trees, high aesthetic value, biodiversity, refugia, biological cores, rarity, and large landscape size. |
| Promotes landscape connectivity | Enhances and/or does not impair landscape connectivity in the context of wildlife species support, such as corridors for migration. |
| Protects water quality | Protects and/or does not impair water quality |
| Promotes accessible recreational opportunities | Promote consideration of strategies in areas that are accessible to visitors and/or provide recreational opportunities, including public lands. |
| Enhances or protects cultural/heritage values | Dedicate careful consideration of adaptation strategies in areas with human connections to landscape of regional inhabitants, including tribal concerns and heritage values, and ecosystem services related to cultural or spiritual connection to landscape. |
| Reduces flood risk | Promote consideration of strategies with focus on preservation and/or restoration of floodplain function and ecosystem services. |

Beyond changes in seasonality, hydroclimate modeling output metrics that focused on changes to average and extreme conditions based on the potential to affect ecosystems (Table 2) were requested by the environmental subgroup. Extreme precipitation can influence regional ecology of ephemeral streams characteristic of dryland regions by altering

energy and mass transport dynamics and removing streamside vegetation. Extreme precipitation can also provide opportunities to support riparian health by restoring groundwater levels and encouraging new generations of riparian plant seedlings to be established [49–51]. Extreme or prolonged heat can increase evapotranspiration demand of plants, reduce soil moisture and in extreme conditions can lead to large-scale die-offs [52–54]. Vapor-pressure deficits during droughts rise exponentially with temperature, challenging plants to either shut down growth to conserve water, or risk wilting [55], a factor which led to widespread loss of gross primary productivity and reduced carbon uptake in the Southwestern US during summer 2020 [56].

**Table 2.** Requested hydroclimate metrics from environmental stakeholders; seasonal metrics were requested for evaluation across both climate scenarios.

| Projections | Summary Metrics |
| --- | --- |
| Precipitation | Basin average; top 10% |
| Temperature | Basin average; top 10%, bottom 10% |
| Streamflow | Predicted change in annual runoff at various concentration points; fractional change; top 5 increase in no-flow days; top 5 changes in soil moisture |

## 3. Results

### 3.1. Hydroclimate Modeling with Environmental Implications

Hydroclimate modeling results are summarized in this section in the context of their relevance to the interests of the environmental subgroup; as noted previously, modeling approaches were tailored for specific metrics according to environmental stakeholders' requests. Consistent with most climate projections, our modeling results included an overall increase in temperature across climate scenarios, with more variable implications for changes in precipitation. The choice of downscaling methods (statistical versus dynamic) appeared to influence the range of precipitation projections particularly in the context of larger, extreme events, which were more prevalent in the "worse case" scenario that employed a dynamic downscaling methodology (Figure 6a,b; Table 3a,b). Precipitation projections developed with the dynamically downscaled climate scenario produced a larger "tail" than the statistically downscaled climate scenario; this has been attributed to the statistically downscaled model being tied to historical envelopes of precipitation distributions [57]. This discrepancy in distributions between downscaling approaches is partially attributed to the assumption of stationarity in the statistically downscaled scenario, which is avoided in the dynamically downscaled scenario.

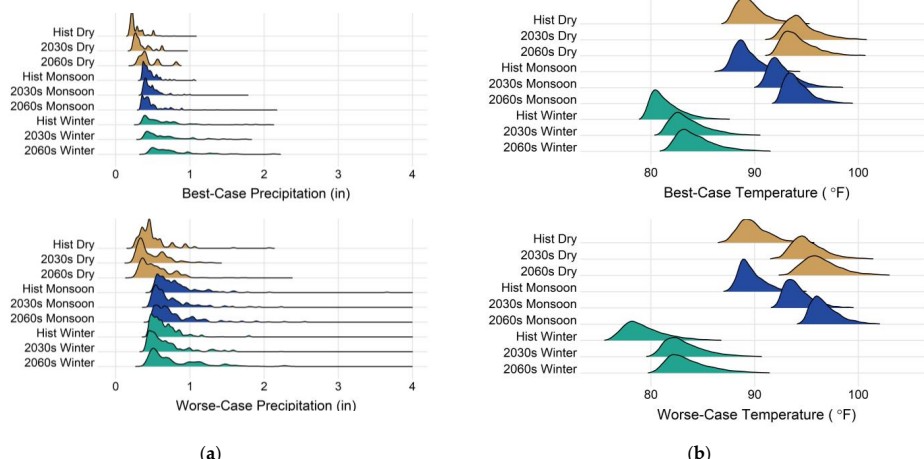

**Figure 6.** Results of (**a**) precipitation projections and (**b**) temperature projections from hydroclimate modeling efforts, summarized by climate scenario. Printed with permission from the US Bureau of Reclamation.

**Table 3.** Summaries of projected (a) temperature and (b) precipitation from 30-year average (1970–1999), averaged over project area. Summaries of (a) precipitation projections and (b) temperature projections from hydroclimate modeling efforts, summarized by climate scenario. Printed with permission from the US Bureau of Reclamation.

| (a) | | | | |
|---|---|---|---|---|
| | Best Case—2030s "Near Future" | Best Case—2060s "Far Future" | Worse Case—2030s "Near Future" | Worse Case—2060s "Far Future" |
| Change in average annual temperature | 2.94° F | 3.83° F | 3.41° F | 5.12° F |
| Change in average dry season temperature | 2.59° F | 2.31° F | 3.44° F | 3.34° F |
| Change in average monsoon temperature | 1.96° F | 3.52° F | 4.24° F | 5.81° F |
| Change in average winter temperature | 1.88° F | 1.85° F | 2.45° F | 3.20° F |
| (b) | | | | |
| | Best Case—2030s "Near Future" | Best Case—2060s "Far Future" | Worse Case—2030s "Near Future" | Worse Case—2060s "Far Future" |
| Change in total annual precipitation | 0.32″ | −0.85″ | −4.34″ | −3.90″ |
| Change in average monsoon precipitation | 0.80″ | −0.87″ | −2.38″ | −1.57″ |
| Change in average winter precipitation | −0.21″ | 0.57″ | −2.25″ | −2.38″ |

Stakeholder requests for hydroclimate modeling outputs included the locations where riparian areas would be most impacted in terms of water supply (e.g., low flow days, increase in dry days). Sub-basins located in Sabino Creek and Cienega Creek were projected to be highly impacted in the context of the greatest increases in days with no streamflow and decreases in soil moisture, which can greatly impact plant and wildlife species that depend on intermittent or perennial streamflow. It is also notable that these locations are located in higher elevations that generally have more flow permanence and higher soil moisture than lower elevation desert basins.

Results were used to identify areas of concern and to prepare for a series of workshops focused on adaptation strategy development and evaluation. Environmental objectives, hydroclimate modeling outputs, and previous restoration projects were compiled to support the development and evaluation of adaptation strategies focused on environmental benefits in the context of water resource imbalances.

### 3.2. Development and Evaluation of Adaptation Strategies

Adaptation strategies were developed by project stakeholders in order to address and/or mitigate the effects of water resource imbalances identified through multi-step modeling conducted as part of the LSCRBS. Technical information developed in this study via modeling efforts was integrated with stakeholder experience and professional knowledge via two workshops that were held to brainstorm and refine adaptation strategies to address projected water resource imbalances. Areas of environmental concern were integrated with additional information (e.g., current and projected groundwater depletion) to identify priority areas of concern across the study area. Areas of concern that included environmental impacts include the Canada del Oro (CDO) and Sabino Creek-Tanque Verde (SC-TV) (Areas 1 and 2 in Figure 7).

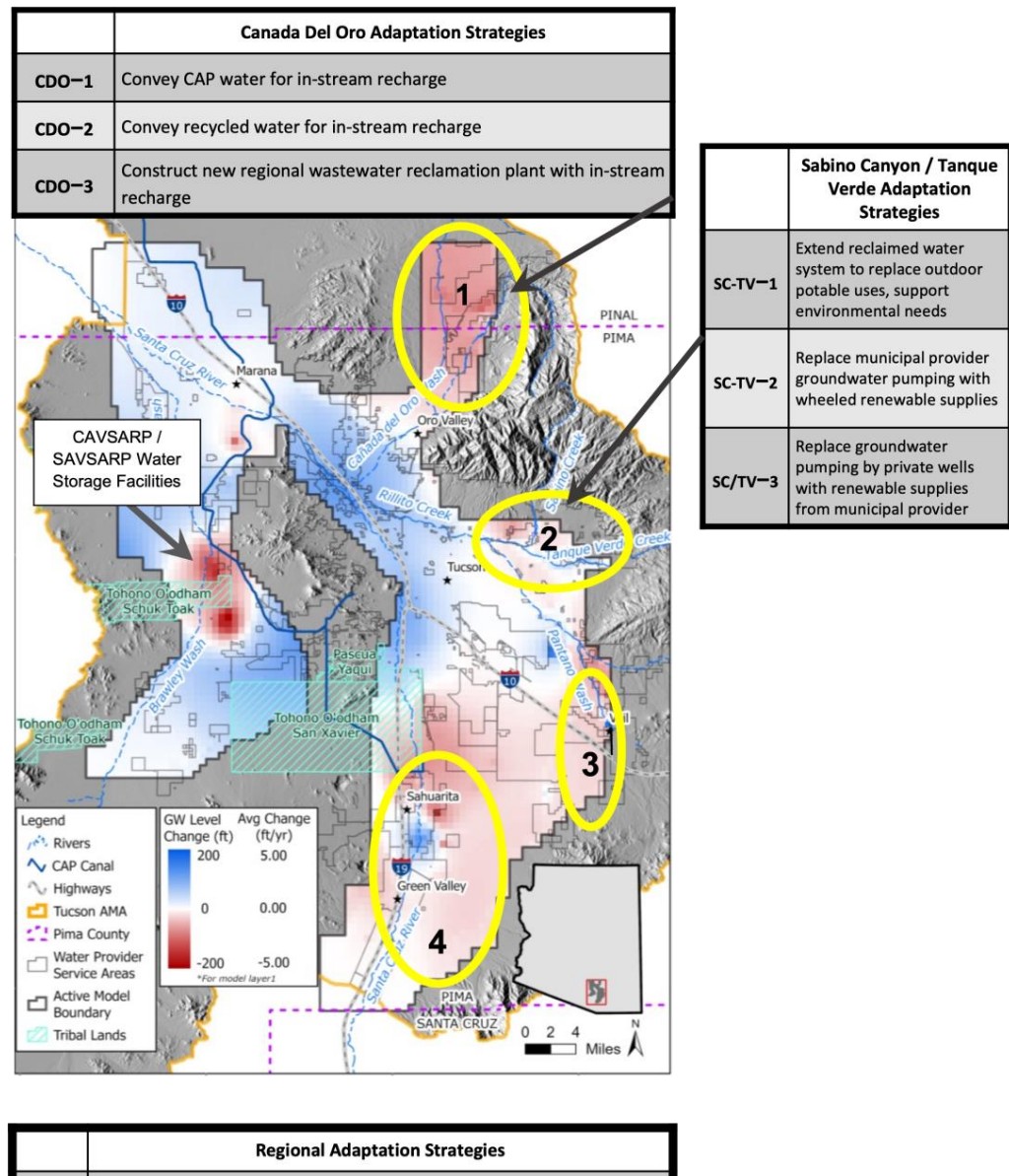

**Figure 7.** Map of areas of concern identified in LSCRBS, presented on a map of groundwater modeling results developed in later-phase study efforts, with environmental adaptation strategies presented in the proposed region (where applicable). Areas of concern identified in the LSCRBS include (1) Canada del Oro, (2) Sabino Canyon/Tanque Verde, (3) Southeast Tucson, and (4) Green Valley. Additional notable areas in the study area include the Southern Avra Valley Storage and Recovery Project (SAVSARP) and the Central Avra Valley Storage and Recovery Project (CAVSARP) due to their role in supporting regional water resource demands.

Adaptation strategy approaches and outcomes are summarized where related to environmental adaptation strategies (Table 4). Adaptation strategies in the CDO region were primarily centered around in-stream recharge, which can provide direct benefits to local riparian systems, while strategies proposed in the SC-TV area were primarily focused on offsetting local groundwater pumping to encourage shallow groundwater tables to support local riparian ecosystems. Strategies that incorporated stormwater for

use were considered on a regional (i.e., not site-specific) basis that ranged across scales of implementation, labeled with the "REG" prefix in Table 4.

**Table 4.** Adaptation strategies that included environmental benefits in the study.

| Strategy Acronym | Strategy Name | Description | Environmental Benefits |
|---|---|---|---|
| CDO-1 | CAP water to CDO area with in-stream recharge | Construct pump stations and a pipeline to convey water from the Central Arizona Project (CAP) Red Rock Pumping Plant to the Cañada del Oro (CDO) wash area for in-stream recharge. | In-stream recharge; riparian health; promote recreation |
| CDO-2 | SCR reclaimed to CDO area with in-stream recharge | Construct pump stations and a pipeline to convey reclaimed water in the Santa Cruz River (SCR) past Trico Road to the Cañada del Oro (CDO) wash area for in-stream recharge. Includes treatment for PFAS and 1,4 dioxane. | In-stream recharge; riparian health; promote recreation |
| CDO-3 | Saddlebrooke area sub-regional WRF with in-stream recharge | Construct a new sub-regional wastewater reclamation facility (WRF) in the Saddlebrooke area with a pipeline for in-stream recharge. | In-stream recharge; riparian health; promote recreation |
| SC-TV-1 | Tucson Water RECLAIMED SYSTEM EXTENSION to Isabella Lee Preserve and other sites | Extend Tucson Water (TW) reclaimed water pipeline to replace outdoor use of potable water at new sites and support irrigation at Isabella Lee Preserve. | Reduce groundwater overdraft; riparian health; promote recreation |
| SC-TV-2 | Wheel Tucson Water renewable supplies to metro water district hub service area | Use existing and new infrastructure to connect the Metro Water District's Hub Service Area to the Tucson Water central potable system. Wheel renewable supplies, replace need for local groundwater pumping. | Reduce groundwater overdraft; riparian health |
| SC-TV-3 | Tucson Water renewable supplies to exempt well owners | Connect exempt well owners to Tucson Water central potable system to provide renewable supplies, retire exempt wells. | Reduce groundwater overdraft; riparian health; promote recreation |
| REG-1 | Stormwater management using low-impact development | Retrofit existing properties and incorporate low-impact development (LID) features into new developments to harvest and use stormwater at the residential scale. Build-out takes place from 2020–2060. | Flood control; terrestrial ecosystem health |
| REG-2 | Stormwater multi-purpose, multi-use basins and channels | Construct large multi-benefit retention and detention basins and channels to collect and slow stormwater from impervious areas for flood mitigation, storage and/or habitat restoration | Flood control; terrestrial ecosystem health |
| REG-3 | Stormwater management using upland restoration | Perform upland restoration using small, distributed features to restore, protect and enhance surface water resources. May provide improved infiltration and soil moisture and restore groundwater levels. | In-stream recharge; riparian health; promote recreation |

Strategies were refined and evaluated via an economic and tradeoff analysis process in later phases of the basin study. Due to the diversity of stakeholder interests and objectives, adaptation strategies were presented in terms of "menu options" that were ranked in several different ways, but no specific recommendations were made as a result of this study (consistent with the original objectives of the study, which were to identify locations of supply and demand imbalance, but not to identify a specific path forward).

## 4. Discussion

The LSCR Basin Study developed a place-based assessment of supply and demand imbalances by integrating regional climate projections and stakeholder-generated data and knowledge. It also provides an example of the 2009 SECURE Water Act being used in a manner that considers water supply reliability in the context of environmental concerns, resulting in the generation of adaptation strategies that integrate desirable environmental and water supply outcomes. Processes used to engage study partners to generate adaptation strategies were designed to be highly interactive and contributed to the generation of strategies that were supported by a broad range of stakeholders.

There are many methods described in the scientific literature and/or used by practitioners to assess the influence of climate change and variability on water resources, e.g., [43,58–60]. There is limited literature, however, on engagement of water management interests in the

development of adaptation options. These efforts are often conducted without gathering regional stakeholder or user input or true co-production of knowledge [46,61–63]. Regional climate knowledge that is agreed-upon and perceived as legitimate is key to establishing a strong foundation for dialogue between scientists and a range of stakeholders, and can have a strong influence upon long-term regional resilience [13,46,64]. Co-produced climate information in this study was developed with the goals of increasing the perceived saliency and applicability of hydroclimate modeling results. An example of this effort was the request from stakeholders for hydroclimate model outputs that could be applied to the assessment of potential environmental impacts under multiple scenarios (e.g., changes in growing season as a result of projected temperature changes) [5,65].

Overall, the development of place-based hydroclimate information in partnership with stakeholders resulted in outputs that were considered useful by the study partners as they identified the locations of the highest potential impacts of climate change across the study area both from a water supply and a riparian protection perspective. Of particular interest was information related to changes in the length of the "dry season," including changes in the length of dry periods, and changes related to extreme precipitation. The evaluation of scenarios from an environmental perspective allowed for consideration of impacts on riparian areas and other key environmental resources. It is useful to identify riparian-related areas likely to experience the highest degree of impacts and assessment of changes in precipitation and temperature that can affect environmental habitat suitability for species across the study region. This approach can be applied to environmental management and planning efforts in other areas in which wet/dry seasonality and/or highly variable rainfall and streamflow strongly influence environmental functions [6,66].

Resource planning efforts such as the LSCRBS provide an avenue to integrate hydroclimate forecasts with stakeholder priorities to develop and implement strategies that will help meet current and future water demands [67]. The evaluation of scenarios from an environmental perspective allowed for consideration of impacts to riparian areas and other key environmental resources. This evaluation is significant in the context of regional water resource planning, where water needs for urban areas and industries have traditionally been the primary focus without consideration of environmental water needs [67–69]. The integration of environmental concerns into the overall LSCRBS methodology represents progress towards a larger paradigm shift that considers environmental needs as part of water resource concerns. Stakeholders engaged as part of the LSCRBS found value in using the study process to identify riparian-related areas likely to experience the highest degree of impacts and assessment of changes in precipitation and temperature that can affect environmental habitat suitability for species across the study region.

Despite documented impacts of historical water management decisions on riparian ecosystems, there are opportunities to mitigate negative impacts, and potentially improve riparian health and function, by adopting strategies that consider and prioritize ecosystem health [27,65,68]. Stakeholders suggested the use of "alternative sources" of water such as reclaimed water, effluent, and stormwater to support riparian health in addition to reduction of pumping in private groundwater wells in shallow groundwater areas. These strategies build upon regional efforts to use non-potable reuse options to support environmental and ecosystem health [70–72]. While environmentally-focused adaptation strategies were generally considered on a regional scale, it is our hope that this work will provide another example of environmental priorities being incorporated in planning studies and lead to further development of environmental enhancement projects.

## 5. Conclusions

We present a case study of a hydroclimate modeling effort that incorporated co-production of climate information applicable to environmental management concerns prioritized by stakeholders of the LSCRBS. We conducted a multi-step hydroclimate modeling and risk assessment process to assess a range of future scenarios for the Lower Santa Cruz River Basin in the greater Tucson, Arizona watershed. This work was conducted in

the context of a partnership between the US Bureau of Reclamation and a wide array of water interests and sectors, pursuant to the 2009 SECURE Water Act. Our work included projections of temperature, precipitation, runoff, soil moisture, groundwater recharge, and evapotranspiration, with a particular focus on implications for aquatic habitat and ecosystem health in riparian areas.

Key hydroclimate modeling process decisions were informed by ongoing multi-stakeholder engagement. The LSRBS included a strategic approach to inclusion of stakeholders, resulting in a high level of stakeholder participation from multiple sectors over a period of more than five years. In addition to sharing data and on-the-ground knowledge, the project participants directly influenced the overall project approach, the selection of adaptation options, and the criteria used to evaluate the alternatives. The environmental subgroup played a substantial role in helping to connect the basin study outcomes to issues related to management and design of adaptation options for protection of environmental resources as climate change impacts advance. To our knowledge, the focus on inclusion of environmental protection priorities into the modeling decisions and design process has not been a priority in other reclamation basin studies, and literature in this area is limited. Our study shows that careful selection of global climate models downscaled to develop locally-relevant climate indicators along with linked surface water and groundwater modeling can lead directly to an understanding of local supply and demand imbalances that is useful in a riparian management context and can help with the development of adaptation options to preserve environmental assets.

Future efforts in the region are expected to include more in-depth, design-oriented studies generated by project partners to further assess the utility of strategies identified in the LSCRBS and collaborative discussions to identify specific riparian reaches to focus recharge efforts in the Lower Santa Cruz River Basin. To promote long-term accessibility and usability of technical information developed as part of the LSCRBS, a web-based GIS tool was developed to summarize hydroclimate modeling results across sub-basins for stakeholders to aid in data access and to inform these in-depth studies while informing a range of future water resource management planning and prioritization efforts.

**Author Contributions:** Conceptualization, N.G., L.B., K.J., E.H., C.C., H.-I.C. and J.F.; Data curation, C.C. and H.-I.C.; Formal analysis, L.B., C.C. and H.-I.C.; Funding acquisition, E.H.; Investigation, N.G., L.B., K.J., E.H., C.C. and H.-I.C.; Methodology, N.G., L.B., K.J., E.H., C.C., H.-I.C. and J.F.; Project administration, K.J., E.H. and C.C.; Resources, N.G., L.B., K.J., E.H., C.C., H.-I.C. and J.F.; Software, L.B., C.C. and H.-I.C.; Supervision, K.J., E.H. and C.C.; Validation, L.B. and C.C.; Visualization, L.B. and E.H.; Writing—original draft, N.G., L.B., K.J., E.H., C.C., H.-I.C. and J.F.; Writing—review and editing, N.G., K.J., E.H., C.C., H.-I.C. and J.F. All authors have read and agreed to the published version of the manuscript.

**Funding:** This research was funded by the Bureau of Reclamation and non-federal partners under Memorandum of Agreement 16-MOA-32-0010. Hydroclimate modeling conducted by University of Arizona was funded under grant agreement number R17AP00061, titled, "Dynamically Downscaled Climate Projections in the Lower Santa Cruz River Basin Study."

**Data Availability Statement:** The data presented in this study are available in the supplemental report of Bureau of Reclamation technical memorandum No. ENV-2021-035.

**Acknowledgments:** The authors would like to acknowledge the participation of study partners that include the Southern Arizona Water Users Association (SAWUA), Arizona Department of Water Resources (ADWR), Central Arizona Project (CAP), Pima Association of Governments (PAG), and the Cortaro-Marana Irrigation District.

**Conflicts of Interest:** The authors declare no conflict of interest.

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
