# Peer review of "Stakeholder-Informed Hydroclimate Scenario Modeling in the Lower Santa Cruz River Basin for Water Resource Management"

_water, doi:10.3390/w15101884_

Round 1

Reviewer 1 Report

The article is titled ‘Stakeholder-Informed Hydroclimate Scenario Modeling in the Lower Santa Cruz River Basin for Water Resource Management’. The authors conducted a multi-stage hydroclimate modeling and risk assessment process to assess a range of future changes, with particular emphasis on impacts on ecosystem health. In my opinion, the topic is interesting, but the presentation is unclear and the structure of the article is incorrect. Some issues are described in too general a way. My comments are as follows:

- complete the introduction, indicate the state of research on the analyzed issue and refer to the literature,

- clearly state the aim of the paper, indicate the gaps in the existing literature, which is the novelty of this paper

- section 1.1 and 1.2 should be included in the methodological part. The study area was not well described, the reader did not receive sufficient knowledge about the problem and the current water needs of various stakeholders

- the methodology adopted in this article is not sufficiently well described. The authors did not present the sequence and scope of activities, as well as the methodology used. I propose to present the methodologies in the form of a diagram. The authors in line 137 indicated that they carried out a number of stakeholder engagement activities without indicating what those activities were, what was the group of stakeholders and what arrangements were made

- the article lacks conclusions

Technical Notes:

- adapt the record of literature to the requirements of the journal

- figures are illegible, they should be corrected

- all names used in the article should be placed on the map

Reviewer 2 Report

The paper is very interesting,  raises the issues of impacts of climate change, land use and population growth on projected water supply and develops adaptation strategies for the next 40 years. The hydroclimate modeling process decisions were informed by ongoing multi-stakeholder engagement, which is very valuable and rare.

However, there is no reference to groundwater resources, has the impact of the above factors on changes in groundwater resources been analysed? Do any stakeholders use the groundwater? Have any models been developed in this area?

Figure 2 - Conceptual Diagram of Precipitation-based Seasons - is very simple, please provide some details about the climate parameters for the specified seasons.

Round 2

Reviewer 1 Report

The authors made a partial correction of the article. In my opinion, the discussion needs improvement- the authors referred to 4 items of literature quoted in one sentence.
Technically, the whole article needs to be corrected. The order of the subsections is incorrect, e.g. in chapter 1 there is subsection 1.1 and 1.3, in line 295 there is chapter 21.21, while in line 333 there is chapter 1.2, line 342 is section 2.1. The order of cited literature items is incorrect
